# COVID-19 Vaccine Timing and Co-Administration with Influenza Vaccines in Canada: A Systematic Review with Comparative Insights from G7 Countries

**DOI:** 10.3390/vaccines13070670

**Published:** 2025-06-21

**Authors:** Farah Al hashimi, Sherif Eneye Shuaib, Nicola Luigi Bragazzi, Shengyuan Chen, Jianhong Wu

**Affiliations:** 1Laboratory for Industrial and Applied Mathematics (LIAM), York University, 4700 Keele St., Toronto, ON M3J 1P3, Canada; fhashimi@yorku.ca (F.A.h.); ssherif@yorku.ca (S.E.S.); robertobragazzi@gmail.com (N.L.B.); 2Department of Mathematics and Statistics, York University, 4700 Keele St., Toronto, ON M3J 1P3, Canada; chensy@yorku.ca

**Keywords:** COVID-19, vaccination timing, booster strategies, co-administration, seasonal vaccines, public health interventions, respiratory disease prevention

## Abstract

Background/Objectives: Despite significant advancements in vaccine development and distribution, the optimal timing and integration of COVID-19 vaccination in Canada remain crucial to public health. As the SARS-CoV-2 virus continues to evolve, determining effective timing strategies for booster doses is necessary to sustain immunity, especially in high-risk populations. This systematic review aims to critically evaluate the timing and co-administration strategies of COVID-19 vaccines in Canada, comparing them with approaches in other G7 nations. Methods: The review seeks to identify best practices to inform national vaccination policies, with a particular focus on synchronizing COVID-19 and seasonal influenza vaccinations. We systematically searched Scopus, PubMed, Medline, and Web of Science (17 August 2021 to 7 July 2024) using the PECOS framework. Two independent reviewers screened titles/abstracts, extracted key data on immunogenicity, efficacy, and safety, and performed a narrative synthesis on timing and co-administration outcomes. Results: Evidence summarized across G7 countries reveals that most nations are converging on annual or flexible booster schedules tailored to high-risk groups, often aligning COVID-19 vaccination with influenza campaigns. Countries like Canada, the UK, and the US have integrated these efforts, while others maintain more independent or heterogeneous approaches. In addition, timely booster doses, whether administered annually or more frequently in high-risk settings, consistently reduce infection rates and hospitalizations. Conclusions: These findings collectively support the continued evolution of COVID-19 vaccination programs toward integrated, seasonally aligned strategies. Future public health efforts can build on these lessons not only to sustain protection against SARS-CoV-2 but also to strengthen preparedness for other respiratory infections.

## 1. Introduction

The COVID-19 pandemic has undergone considerable changes since the initial case was publicly reported in Wuhan, China, in December 2019 [1]. As of the release date on 20 July 2024, Canada has reported a total of 4,562,902 COVID-19 cases. Of these cases, 297,583 required hospitalization, while 37,564 were admitted to intensive care units. The total number of COVID-19-related deaths stands at 38,340 [2]. COVID-19 continues to pose a substantial health burden, remaining the primary respiratory infectious disease responsible for hospitalizations and fatalities across all age groups regardless of comorbidity status.

As the COVID-19 pandemic has transitioned into an endemic phase, countries worldwide continue to refine their vaccination strategies, particularly concerning booster dose timing. Among high-income nations, the G7 countries (Canada, France, Germany, Italy, Japan, the United Kingdom, and the United States) provide a valuable comparative framework due to their robust healthcare infrastructures and similar economic profiles. However, these countries also exhibit notable differences. Some countries have some of the oldest populations globally, increasing their vulnerability to severe COVID-19 outcomes [3]. The United States experienced some of the highest case and mortality rates, while other G7 countries maintained relatively lower rates early in the pandemic through targeted public health measures [4], and the implementation of varying containment policies. In addition, differences in healthcare system structures, vaccine procurement strategies, and public trust in health institutions have led to varied levels of vaccine uptake and booster acceptance across the G7 countries [5]. Comparative analyses of these strategies are essential for identifying best practices and informing adaptive policies that can respond to challenges such as waning immunity and emerging variants. Notably, the G7 Health Ministers have emphasized the importance of coordinated efforts in pandemic preparedness and response, as reflected in their 2022 communiqué [6].

In Canada, the impact of COVID-19 vaccination efforts has been profoundly beneficial. According to a study by Wyonch and Zhang [7], from January 2021 to May 2022, vaccinations effectively reduced the number of COVID-19 cases by about 21 percent and hospitalizations by approximately 37 percent. Most notably, for Canadians over 50 years old who are at the highest risk of severe outcomes from the virus, the vaccination efforts have been particularly impactful, reducing mortality by an estimated 63 percent, which translates to about 34,900 lives saved. Additionally, the vaccinations have helped avert approximately 65,000 cases of long-term COVID-19, further underscoring the critical role of these public health measures in mitigating the pandemic’s burden. Public acceptance of COVID-19 vaccines was notably high upon their introduction. As of 15 September 2023, COVID-19 vaccination coverage in Canada reported high rates among the population. A total of 83.2% of Canadians had received at least one dose of a COVID-19 vaccine, and 80.5% had completed the primary schedule [8]. These statistics highlight the strong public engagement with the vaccination program in Canada. However, as the virus continued to mutate, additional booster doses and variant-specific vaccines were developed to better shield the population from the rapidly evolving SARS-CoV-2 virus. Nevertheless, the uptake of subsequent doses has been considerably lower than the initial doses [9]. This trend reflects the challenges in maintaining high vaccination coverage amidst evolving public health dynamics.

COVID-19 vaccination strategies continue to adapt globally in response to the disease’s established endemic status. In Canada, policymakers are navigating complex decisions regarding the optimal timing for COVID-19 booster doses. Although the virus’s threat to severe public health outcomes has diminished compared to the early stages of the pandemic, the potential for future outbreaks underscores the need for well-timed vaccination strategies to mitigate risks, especially for high-risk populations. With reduced COVID-19 lethality, a more precise approach to vaccination timing is essential to balance public health benefits, resource allocation, and public adherence. In response, health authorities, including the World Health Organization (WHO), the National Advisory Committee on Immunization (NACI), and Public Health Ontario, have recently begun examining the feasibility of synchronizing COVID-19 boosters with seasonal influenza vaccines. Research supports that co-administration of COVID-19 and seasonal influenza vaccines can be both safe and effective, particularly for high-risk populations, and offers the practical benefit of reducing the frequency of healthcare visits, which is essential, as both COVID-19 and influenza continue to circulate seasonally. However, despite these insights, the optimal timing and scheduling approach for COVID-19 vaccinations remains an area of ongoing debate and adaptation in Canada [10].

The Public Health Emergency (PHE) status was lifted on 11 May 2023. Although COVID-19 is no longer classified as a public health emergency, it persists as the leading respiratory infectious disease and remains a continuous public health concern, partly due to the emergence of variants with greater transmissibility and severity. On 11 September 2023, a new vaccine targeting the predominant variant at the time, XBB.1.5, received approval. Subsequently, on 12 September 2023, the Advisory Committee on Immunization Practices (ACIP) recommended that everyone aged 6 months and older should receive at least one dose of the 2023/24 COVID-19 vaccine [11]. Despite this clear guidance, as of 30 June 2024, only 18.2% of the population in Canada had received the XBB.1.5 vaccine, with even lower coverage among certain groups. Specifically, vaccination rates were 8.4% among children aged 0 to 4 years, with only 4.5% receiving the XBB.1.5 vaccine [12]. In Canada, ACIP recommendations [11] are not formally adopted or used in policymaking. All vaccine-related guidance in Canada is issued by the National Advisory Committee on Immunization (NACI), which independently assesses scientific evidence and provides recommendations tailored to the Canadian context. While Canadian public health officials may consider ACIP guidance informally particularly when similar vaccine products are used, Canada’s immunization policy is based solely on NACI recommendations.

Recent work by Boikos et al. [13] provides a global overview of co-administration strategies, highlighting the safety, immunogenicity, and increasing uptake of simultaneous BNT162b2 COVID-19 and influenza vaccinations. Their findings support the feasibility of integrated vaccination delivery, particularly in adult populations. However, their analysis primarily focuses on co-administration clinical outcomes and global prevalence. In contrast, our systematic review explores how national vaccination programs, particularly in Canada and other G7 countries, have evolved in timing, scheduling, and policy alignment with seasonal influenza campaigns. This review focuses on timing strategies and country-specific implementation to complement and extend the understanding of how COVID-19 vaccination timing is optimized in high-income settings.

To better understand Canada’s options, this systematic review examines vaccination timing practices in other G7 nations, aiming to identify trends, gaps, and potential best practices in scheduling COVID-19 doses. By systematically examining these approaches, this review aims to generate insights applicable to Canada and other countries striving to maintain high vaccine coverage and public trust. Additionally, by synthesizing insights from international approaches and recent studies, this review aims to provide evidence-informed recommendations to guide Canadian policy decisions on adapting COVID-19 and influenza vaccination schedules to current health risks, including the potential for co-administration of these vaccines.

Recent data from the World Health Organization (WHO) further support this need: a 2024 study from the WHO Regional Office for Europe reported that COVID-19 vaccines prevented over half of COVID-19-related hospitalizations and severe outcomes among high-risk populations, reinforcing the continued value of timely booster doses [14]. In addition, a study by Meslé et al. [15] estimated that at least 1.6 million lives were directly saved by COVID-19 vaccination programs between December 2020 and March 2023 in the WHO European Region, highlighting the real-world impact of well-timed and inclusive vaccine rollouts [15].

## 2. Materials and Methods

### 2.1. Search Strategy

Electronic databases including Scopus, PubMed, Medline, and Web of Science (WOS) were searched systematically for potential publications from 17 August 2021 to 7 July 2024. MedLine was accessed through the Web of Science interface. To ensure the efficacy of our search strategy and the relevance of included studies, our criteria for selection were based on the topics of COVID-19, booster vaccines, and optimal timing. Although ‘co-administration’ was not explicitly used as a search term, relevant studies discussing this concept were captured via general vaccination keywords and manual reference checks.

To search the mentioned databases, we used targeted keywords combined with Medical Subject Headings (MeSH) for PubMed and Medline, using Boolean operators “AND/OR” for combinations. The search was customized for two specific cases: the first focused on Canada, while the second encompassed other G7 member countries (France, Germany, Italy, Japan, the United Kingdom, and the United States).

The search was limited to English-language studies, conference papers, articles, early access materials, and data papers. No gray literature was examined. Table 1 and Table 2 provide a summary of the number of articles retrieved from each database, along with the associated queries for both case 1 and case 2. All retrieved articles were compiled and analyzed using EndNote X9.

In addition to our primary database searches, we performed supplementary citation searches by reviewing the reference lists of key included studies (backward citation searching) and identifying subsequent articles that cited them (forward citation searching) using tools such as Google Scholar. This approach is consistent with PRISMA recommendations and was undertaken to enhance the completeness of our evidence retrieval. Six additional studies were identified through this process and assessed against the inclusion criteria.

### 2.2. Eligibility Criteria

As part of our systematic review, we structured the inclusion and exclusion criteria using the PECOS framework: Population, Exposure, Comparator, Outcomes, and Study Design. PECOS is commonly used in reviews of observational studies to examine the effects of exposures rather than interventions.

In our context, the “Exposure” was defined as receipt of a COVID-19 booster dose, including variables related to timing and co-administration with influenza vaccines. The “Comparator” encompassed differences such as age groups, vaccine types, dosing intervals, and population characteristics. Due to the diversity of study designs and outcome measures across the included studies, we employed a thematic grouping strategy (e.g., by population, vaccination strategy, and outcome type) to facilitate meaningful narrative synthesis. This allowed us to synthesize findings despite the heterogeneity in comparators and outcomes, as suggested in the SWiM (Synthesis Without Meta-analysis) reporting guidelines [16].

The full inclusion and exclusion criteria, as guided by PECOS framework, are presented in Table 3.

### 2.3. Study Selection

The study selection was carried out independently by two reviewers. The reviewers screened all articles obtained from the database (after removing duplicates) initially by title and abstract, followed by full-text screening. The eligibility criteria were applied during the screening process. In cases of disagreements between the first two authors, a third author provided input. For publications lacking an abstract or when the specified criteria could not be assessed from the title or abstract, a full-text assessment was performed. During the second phase, relevant information was analyzed to determine if the study aligned with the proposed research question.

### 2.4. Data Extraction

Data were extracted from all eligible papers using a predefined electronic form containing the following components: (1) the characteristics of the included studies (reference, publication year, title); (2) study type; (3) population considered; and (4) study outputs.

### 2.5. Narrative Heterogeneity Assessment

While a formal statistical heterogeneity analysis (e.g., I^2^) was not applicable due to the lack of meta-analytical computation, heterogeneity across the included studies was addressed through structured narrative synthesis. Included studies were grouped by country, population type, exposure timing, co-administration of COVID-19 and influenza vaccine, and outcome category. This method allowed us to explore and report sources of variability across studies in alignment with the SWiM (Synthesis Without Meta-analysis) reporting guideline [16].

## 3. Results

In our systematic search, we identified 3983 records. After removing 565 duplicates, 3418 records remained for the initial screening. During the review of titles and abstracts, 3345 studies were excluded based on the Stage 1 inclusion criteria. Most of the excluded studies focused broadly on vaccine effectiveness or uptake, but did not address the timing of COVID-19 or influenza booster administration, the primary focus of our review. Studies were also excluded for being conducted outside G7 countries or lacking relevant outcomes. As such, specific exclusion reasons were not documented at this stage, in line with standard systematic review practices [17]. Of the 73 studies left for full-text screening in Stage 2, 59 studies did not meet the eligibility criteria outlined in Table 3, resulting in 14 studies being included in the systematic review. Additionally, six studies were identified through citation searches, of which four met the eligibility criteria. This brought the total number of included articles to 18, as shown in Figure 1.

### 3.1. General Characteristics of the Included Studies

A total of 18 studies were included, spanning multiple countries, institutions, and methodological approaches, primarily from the United States, Canada, and parts of Europe. Most studies were conducted by prominent academic or public health institutions such as Yale School of Public Health [18,19], the Centers for Disease Control and Prevention (CDC) [20,21], University of North Carolina [22,23], and Public Health Ontario [24,25] as well as industry-affiliated partners such as Pfizer [26,27] and Moderna [28]. Participant populations varied widely, including real-world individuals (e.g., healthcare personnel, adults aged 50+, nursing home residents [20,23,29]), as well as simulated populations modeled on national demographics [18,30]. The general characteristics of all included studies are summarized in Table 4. Methodologies included retrospective cohort studies [28], test-negative case-control designs [24,25], online cross-sectional surveys [31], and various modeling approaches such as agent-based, dynamic transmission, and microsimulation models [18,27,29,32]. Key themes across the studies included booster dose timing, vaccine safety and effectiveness, co-administration with seasonal influenza vaccines, and economic impact. Several U.S.-based studies found that shorter booster intervals reduced hospitalizations and deaths, particularly among older or high-risk individuals [22,29,32]. Canadian studies highlighted high acceptance for third doses [31], strong booster effectiveness during early post-vaccination periods [24], and waning immunity over time, especially during Omicron subvariant waves [33]. European studies contributed additional data on seasonality and concurrent vaccine responses. Studies focusing on the co-administration of COVID-19 and influenza vaccines reported favorable safety outcomes and, in some cases, enhanced immunogenicity when vaccines were administered in separate arms [34,35]. Overall, the studies suggest that strategically timed booster campaigns can optimize public health outcomes, especially when aligned with flu season [27].

### 3.2. Impact of Vaccine Dosing Intervals and Administration Strategies

Table 5 provides a comparative analysis of various COVID-19 vaccination strategies across G7 countries, focusing on dosing intervals, their effectiveness in reducing infection rates and hospitalizations, and associated safety outcomes. The studies collectively indicate that synchronized COVID-19 and influenza vaccinations, as well as appropriately timed booster doses, can enhance protection against COVID-19, reduce healthcare utilization, and maintain acceptable safety profiles. This underscores the importance of strategic vaccination planning to optimize public health outcomes.

### 3.3. Integration of COVID-19 Vaccination with Other Respiratory Vaccines

Table 6 presents an overview of COVID-19 and influenza vaccine co-administration strategies across several G7 countries. Most countries employed same-day or concurrent administration, particularly targeting older adults and high-risk groups. Key findings highlight that co-administration generally maintains safety, reduces hospitalizations, and improves public acceptance. While most studies reported no unexpected adverse effects, some noted mild side effects when vaccines were administered together. Overall, the data supports COVID-19 and influenza vaccine co-administration as an efficient and acceptable strategy for managing respiratory infections.

## 4. Discussion

COVID-19 vaccination strategies in Canada and other G7 nations reveal both similarities and distinctions, particularly in terms of vaccine types, dosing schedules, and integration with other seasonal vaccines like influenza. These variations highlight key policy choices and their respective public health impacts. Canada has shown a proactive stance by considering COVID-19 and influenza vaccine integration to ease logistics and maximize public health impact. Similar integration strategies are observed across G7 countries, reflecting a coordinated approach to combat COVID-19 as seasonal data and epidemiological patterns indicate potential overlap in peak infection periods.

The impact of COVID-19 vaccine dosing intervals and administration strategies has been extensively studied across G7 countries, including Canada. As summarized in Table 5, key studies have evaluated various approaches to optimize vaccine effectiveness, manage infection rates, and reduce hospitalizations while ensuring safety across different strategies. In Canada, modeling by Fisman et al. [33] assessed booster vaccination at intervals ranging from 2 to 24 months and found that vaccinated populations had consistently lower infection risks compared to unvaccinated groups across all scenarios. Similarly, research in the United States by Stoddard et al. [30], Park et al. [32], and Lin et al. [22] supports more frequent boosters (e.g., every 3–6 months or semi-annually), particularly among older or high-risk groups, indicating improved protection against infection and severe outcomes. The co-administration of COVID-19 and influenza vaccines has also been explored. Mehta et al. [28] found that combined vaccination for individuals aged 50 and above resulted in reduced hospitalizations and overall healthcare costs, with a low risk of adverse events. Barouch et al. [35] and Aydillo et al. [34] reported that same-day co-administration maintains strong antibody responses without reducing immunogenicity for either vaccine, while Moro et al. [20] confirmed the absence of unexpected safety signals. Additionally, Wiemken et al. [27] observed that COVID-19 rates peaked during winter, supporting the synchronization of COVID-19 and flu vaccinations. These findings highlight the importance of tailored booster dosing intervals and integrated vaccination strategies to effectively manage COVID-19 across different populations and healthcare settings.

Regarding the co-administration of COVID-19 and influenza vaccines, as shown in Table 6, multiple studies confirm the effectiveness and safety of administering COVID-19 and influenza vaccines in a single visit. Barouch et al. [35] reported that same-day administration of COVID-19 and influenza vaccines enhanced antibody responses without interfering with the immune response to influenza. Moro et al. [20] found that same-day co-administration was well-tolerated, with no unexpected adverse events and only mild side effects such as fatigue. In the United States, Mehta et al. [28] observed reduced hospitalization rates and healthcare costs in individuals aged 50 and above who received both vaccines during the same visit. Aydillo et al. [34] also supported the immunological benefits and public acceptance of same-day co-administration during fall campaigns. In Canada, Reifferscheid et al. [31] reported that older adults and individuals with chronic conditions showed high acceptance of COVID-19 and influenza co-administration, particularly when receiving booster doses. Overall, these findings demonstrate that co-administration strategies are safe, immunologically sound, and positively received by the public, especially among high-risk populations. This supports continued integration of COVID-19 and influenza vaccine campaigns to improve uptake and efficiency in healthcare delivery. Our review highlights the growing shift toward more efficient immunization practice. These findings align with those of Boikos et al. [13], who demonstrated that it is immunologically safe and practically possible to administer COVID-19 and influenza vaccinations at the same time. Thus, incorporating such strategies may enhance uptake and reduce the healthcare burden during peak respiratory seasons. However, while co-administration of COVID-19 and influenza vaccines has been proposed to improve convenience and coverage, there are practical barriers to implementing this strategy across all age groups. The eligible populations for each vaccine do not always align. A seasonal influenza vaccination is commonly recommended for young children and older adults, whereas COVID-19 boosters are often prioritized for adults and high-risk groups. For example, Moro et al. [20] reported that most co-administration cases occurred in older adults, reflecting existing recommendations. Lin et al. [22] found that booster effectiveness and optimal intervals differ by age, with older adults showing greater benefit and clearer policy guidance, while recommendations for younger groups remain limited and evolving. These differences in eligibility and dosing recommendations make synchronized administration logistically challenging and should be carefully considered in future vaccine planning and scheduling policies.

This review focuses on Canada while drawing comparisons with other G7 countries to ensure contextual relevance. We chose the G7 countries because they have broadly comparable healthcare systems, similar access to mRNA-based vaccines, and analogous climatic patterns, all influencing vaccine deployment strategies and timing. These similarities provide a meaningful basis for comparison and help ensure that insights drawn apply to high-income countries with similar public health capacities. While the findings may not be directly generalizable to low- and middle-income countries in Africa, Asia, or other regions of the world, where vaccine platforms, logistics, and seasonal dynamics differ significantly, the methodological approach used in this study can be adapted and applied to those contexts. Future studies may consider expanding to OECD countries in the future to explore broader applicability across high-income settings.

## 5. Conclusions

The findings of this systematic review underscore the importance of precise timing and strategies for co-administration of COVID-19 and influenza vaccines in sustaining immunity across Canada’s diverse populations. As SARS-CoV-2 has transitioned toward endemicity, synchronizing COVID-19 boosters with seasonal influenza vaccines represents a viable approach to improve public adherence and reduce logistical burdens on healthcare systems.

Recent studies have further underscored the real-world effectiveness of COVID-19 vaccination strategies. A 2024 WHO report [14] found that COVID-19 vaccines prevented over half of hospitalizations and severe outcomes among high-risk populations, reinforcing the value of routine annual vaccination for older adults and those with underlying conditions. In parallel, a 2024 study [15] estimated that COVID-19 vaccination programs directly saved at least 1.6 million lives across the WHO European Region between December 2020 and March 2023. These findings support continued investments in vaccination strategies targeting the most vulnerable populations.

Evidence summarized across G7 countries reveals that most nations are converging on annual or flexible booster schedules tailored to high-risk groups, often aligning COVID-19 vaccination with influenza campaigns. Countries such as Canada, the UK, and the US have integrated these efforts, implementing synchronized vaccination strategies to improve public health outcomes and logistical efficiency. In addition, timely booster doses, whether administered annually or more frequently in high-risk settings, consistently reduce infection rates and hospitalizations. Furthermore, co-administration of COVID-19 and influenza vaccines was generally reported to be safe, effective, and increasingly well received by the public, especially among older adults.

While these findings provide important insights into optimal COVID-19 booster strategies, it is important to acknowledge that some conclusions such as the comparative effectiveness of annual versus biannual schedules are derived primarily from simulation-based studies rather than real-world longitudinal data. Although modeling approaches are valuable for projecting potential outcomes, they are inherently limited by their assumptions and lack of empirical validation. This underscores the need for future longitudinal research to confirm and refine these projections, ensuring that policy decisions are grounded in robust, real-world evidence.

This review also has several limitations. Although we aimed to synthesize evidence across all G7 countries, no eligible studies focusing on Japan met our inclusion criteria after full-text screening. As a result, Japan’s real-world COVID-19 booster strategies were not included in the main evidence synthesis. However, recent external evidence suggests that Japan continues to prioritize boosters for older adults. For example, a recent economic evaluation found that COVID-19 booster strategies remain cost-effective for individuals aged 65 and older, even under conservative modeling assumptions [36], reinforcing the importance of tailored booster programs based on country-specific demographic and economic contexts. Although this review offers detailed comparisons across G7 countries, the exclusive focus on high-income nations may limit the applicability of findings to lower-income settings. Differences in healthcare infrastructure, vaccine access, and demographic profiles must be considered when interpreting these results globally. A broader inclusion of studies from diverse regions would be valuable in future reviews.

Overall, these findings collectively support the continued evolution of COVID-19 vaccination programs toward integrated, seasonally aligned strategies. Future public health efforts can build on these lessons not only to sustain protection against SARS-CoV-2 but also to strengthen preparedness for other respiratory infections. By adopting these evidence-based approaches, Canada can more effectively allocate resources, maintain high vaccination rates, and mitigate the impact of COVID-19, advancing both short-term and long-term public health goals. This review supports policy shifts toward structured annual or biannual vaccination schedules; however, given that much of the current evidence stems from modeling extrapolations, the strength of the annual reinforcement pin needs to be confirmed by prospective studies.

## Figures and Tables

**Figure 1 vaccines-13-00670-f001:**
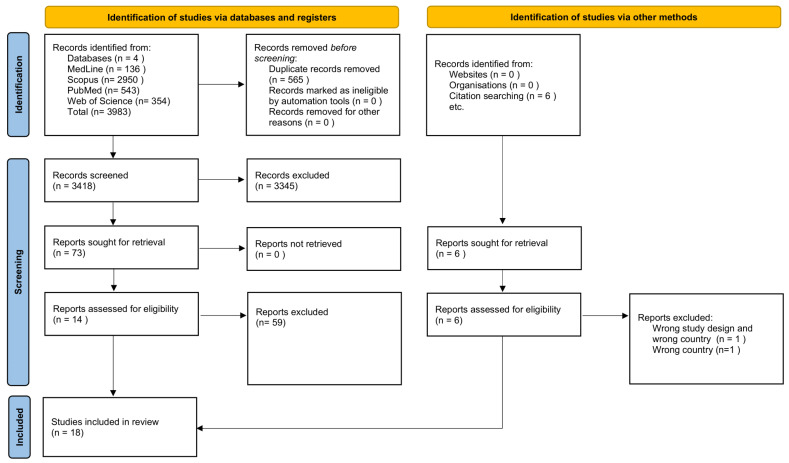
PRISMA Flowchart.

**Table 1 vaccines-13-00670-t001:** Summary of the search results and search queries used for the systematic search, performed on 7 July 2024, for Canada.

Databases	Number of Retrieved Records	Search Terms
Medline	20	See Table A1
Scopus	221	See Table A1
PubMed	54	See Table A1
Web of Science	33	See Table A1
Total	328	

**Table 2 vaccines-13-00670-t002:** Summary of the search results and search queries used for the systematic search, performed on 7 July 2024, for other G7 countries.

Databases	Number of Retrieved Records	Search Terms
Medline	116	See Table A2
Scopus	2729	See Table A2
PubMed	489	See Table A2
Web of Science	321	See Table A2
Total	3655	

**Table 3 vaccines-13-00670-t003:** PECOS Criteria for Eligibility in the Systematic Review.

Criteria	Inclusion Criteria	Exclusion Criteria
Population	Humans	Non-humans, in vitro studies, animal models
Exposure	COVID-19 booster vaccination strategy (e.g., timing, co-administration with influenza)	Studies not focused on booster administration
Comparator	Age group, population type, comorbidities, vaccine type, dosing interval	Irrelevant comparators or no clear comparison
Outcomes	Efficacy, effectiveness, immunogenicity, epidemiologic impact (hospitalizations, infections, disease severity)	Non-epidemiological outcomes or irrelevant endpoints
Setting	Study conducted in Canada or G7 countries (France, Germany, Italy, Japan, United Kingdom, United States)	Study conducted outside of Canada or G7 countries
Study Design	Observational studies (cohort, case-control, cross-sectional) and simulation modeling studies relevant to policy evaluation	RCTs, qualitative, ecological studies, case reports, experimental studies, reviews, congress abstracts, and posters

**Table 4 vaccines-13-00670-t004:** General Characteristics Summary for the Selected Studies.

Authors	Characteristics				
	Institution	Participants	Country	Method	Result
Susanna E. Barouch et al. [35]	Ragon Institute of MGH, MIT, and Harvard; Univ. of Massachusetts at Lowell	Concurrent and separate administration groups	USA	Analyzed IgG1 and neutralization responses	Higher durable SARS-CoV-2 response, no interference with influenza
Emily J. Ciccone et al. [23]	Univ. of North Carolina, Columbia Univ., Duke Univ., and RTI International	Healthcare personnel grouped by serostatus	USA	Blood sampling and antibody testing	Higher initial response in seropositive individuals, persistence up to 9 months
Dan-Yu Lin et al. [22]	University of North Carolina, CDC Foundation	5,769,205 adults, age-based grouping	USA	Linked health data, Cox regression models	Shorter booster intervals reduced hospitalizations and death
Pedro L. Moro et al. [20]	CDC, FDA	3689 participants, diverse age range	USA	Analyzed Vaccine Adverse Event Reporting System	No unexpected adverse events; common side effects included fatigue
Chad R. Wells et al. [18]	Yale School of Public Health	Synthetic population model of U.S. demographics	USA	Age-structured dynamic transmission model	A second dose reduced hospitalizations by 123,869 cases, saving 3.63 billion
Laura Reifferscheid et al. [31]	Univ. of Alberta, BMC Public Health	6010 Canadian adults, national sample	Canada	Online cross-sectional survey with logistic regression	70% acceptance for third dose, co-administration with flu vaccine favored
David N. Fisman et al. [33]	Dalla Lana School of Public Health, Public Health Agency of Canada	Simulated vaccinated and unvaccinated sub-populations	Canada	Compartmental model (SIR) with waning immunity and booster effects	Vaccinated had lower infection risk; contact with unvaccinated increased risk
Timothy L. Wiemken et al. [27]	Pfizer Inc., Columbia Univ., Univ. of Iowa	COVID-19 cases, hospitalizations, and mortality data from U.S. and Europe	USA, UK, France, Germany, Italy	Time series decomposition using Prophet model	Seasonal COVID-19 peaks observed; supports aligning annual boosters with flu season
Timothy L Wiemken et al. [26]	Pfizer Inc.	COVID-19 cases in US and Europe	USA, Uk, France, Germany, Italy	Time series modeling for COVID-19 seasonality analysis	COVID-19 rates peaked during winter respiratory season, supporting annual boosters
Sarah M. Bartsch et al. [29]	CUNY Graduate School of Public Health, California Association of Long Term Care Medicine, Univ. of California Irvine, Johns Hopkins Univ.	Nursing home residents and staff	USA	Agent-based model simulating nursing home COVID-19 spread, vaccination timing, and economic outcomes	Late summer/early fall vaccination was cost-effective, averting 102–105 cases when initiated between July and October
Teresa Aydillo et al. [34]	Icahn School of Medicine at Mount Sinai, University of Seville	128 volunteers receiving influenza and/or COVID-19 vaccines	USA	Antibody and T-cell response quantification in three vaccination groups: flu only, flu and COVID-19 in the same arm, and flu and COVID-19 in different arms	Concomitant vaccination was safe, with higher antibody response when administered in different arms, particularly for H3N2 influenza
Ramandip Grewal et al. [24]	Public Health Ontario, ICES, Univ. of Toronto	Adults aged 50+ in Ontario, Canada	Canada	Test-negative design estimating vaccine effectiveness by time since booster dose	Boosters restored protection against severe outcomes to 92–97% shortly after the dose but waned over time, especially during BA.4/BA.5 Omicron sublineages
Ramandip Grewal et al. [25]	ICES, Univ. of Toronto, McMaster Univ.	Long-term care residents aged 60+ in Ontario, Canada	Canada	Test-negative design comparing fourth vs. third dose effectiveness in long-term care residents	Fourth dose provided additional protection against infection, symptomatic infection, and severe outcomes compared to third dose
Jeffrey P. Townsend et al. [19]	Yale School of Public Health, Georgia Institute of Technology, Univ. of North Carolina at Charlotte	Simulated individuals with prior booster or breakthrough infections	United States	Longitudinal antibody data and projections for optimal booster timing	Suggested delayed boosting after breakthrough infections based on regional infection rates
Sung-mok Jung et al. [21]	Univ. of North Carolina at Chapel Hill, Johns Hopkins Bloomberg School of Public Health, Univ. of Pittsburgh, Pennsylvania State Univ., Northeastern Univ., NIH	Model-based projections with age-based vaccination coverage	USA	Scenario modeling to predict hospitalizations and deaths under different immune escape and vaccination scenarios	Reformulated vaccines significantly reduce morbidity and mortality, especially in high-risk groups during winter
Hailey J. Park et al. [32]	Stanford Univ., Yale School of Public Health, Univ. of California, San Francisco, California Dept. of Public Health	Simulation of age-based and immunocompromised populations	USA	Microsimulation model analyzing effects of booster frequency on severe COVID-19 across age and risk groups	Semiannual boosters benefit older and immunocompromised groups most, with limited impact for younger populations
Darshan Mehta et al. [28]	Moderna Inc.	Adults aged 50+ receiving influenza and/or COVID-19 vaccines	USA	Retrospective cohort study using insurance claims data to evaluate healthcare utilization and costs	Same-day co-administration reduced COVID-19-related hospitalizations and overall medical costs
Madison Stoddard et al. [30]	Fractal Therapeutics, Harvard Medical School, Boston Children’s Hospital	Simulated population with varying vaccination frequencies	USA	Agent-based model for varying COVID-19 booster frequencies and effectiveness	Higher booster frequency showed benefits for population-level COVID-19 control

**Table 5 vaccines-13-00670-t005:** Effectiveness and Safety of Vaccination Strategies Across G7 Countries.

Study	Population (Canada vs. Specific G7 Countries)	Dosing Intervals	Key Findings on Effectiveness (e.g., Infection Rates, Hospitalizations)	Safety Outcomes
Wiemken et al. [27]	USA, UK, France, Germany, Italy	Annual boosters timed with flu season	COVID-19 rates peaked in winter; supporting synchronized COVID-19 and flu vaccination	Safety not assessed
Mehta et al. [28]	USA	Single-dose and combined influenza-COVID-19 vaccination for those 50+	Reduced hospitalizations and overall healthcare costs with combined vaccination	Low risk of adverse events
Stoddard et al. [30]	USA	Frequent boosters modeled (every 3–6 months) for populations with faster antibody waning	Projected to maintain vaccine efficacy and reduce infection and death risk under waning immunity scenarios	Yearly boosters with improved durability projected to reduce variation and enhance protection
Fisman et al. [33]	Canada	Booster vaccination modeled with intervals ranging from 2 to 24 months, including annual schedules	Lower infection rates among vaccinated vs. unvaccinated populations across all scenarios	Not reported
Barouch et al. [35]	USA	Same-day (concurrent) vs different-day COVID-19 and influenza vaccination	Enhanced and sustained spike-specific antibody responses for COVID-19 with concurrent administration; no reduction in influenza response	Safety not assessed in this study
Aydillo et al. [34]	USA	Same-day co-administration of COVID-19 and influenza vaccines	Maintained antibody responses for both vaccines; enhanced response to influenza H3N2 when administered in separate arms	Mild reactogenicity profile; no safety concerns observed
Moro et al. [20]	USA	Same-day co-administration of COVID-19 and influenza vaccines	Effectiveness not assessed	No unexpected safety signals; most reported events were mild (e.g., fatigue, headache)
Park et al. [32]	USA	Semiannual vs. annual vs. one-time boosters in older and immunocompromised populations	Semiannual boosters substantially reduced severe COVID-19 outcomes in high-risk groups; annual boosters showed limited impact for younger adults	Safety outcomes not explicitly modeled; focus was on projected effectiveness
Lin et al. [22]	USA	Booster intervals of 6, 9, and 12 months	Shorter intervals associated with significantly lower risks of infection, hospitalization, and death; strongest effects observed in older adults	Not reported

**Table 6 vaccines-13-00670-t006:** Co-administration of COVID-19 and Influenza Vaccines: Insights and Findings.

Country	Type of Vaccines	Co-Administration Strategy	Key Findings	Authors
USA	COVID-19, Influenza	Same-day vs. different-day administration of COVID-19 and influenza vaccines	Enhanced antibody response; no interference with influenza response	Susanna E. Barouch et al. [35]
USA	COVID-19, Influenza	Same-day administration	No unexpected adverse effects; mild side effects like fatigue	Pedro L. Moro et al. [20]
Canada	COVID-19, Influenza	Public acceptance of co-administration with influenza vaccines	Older adults and individuals with chronic illness showed higher acceptance of both additional COVID-19 doses and co-administration with influenza vaccines	Laura Reifferscheid et al. [31]
USA	COVID-19, Influenza	Same-day co-administration in fall campaigns	Positive public response; antibody response maintained	Teresa Aydillo et al. [34]
USA	COVID-19, Influenza	Same-day co-administration vs. influenza vaccine only (adults 50+)	Lower all-cause and COVID-19–related hospitalizations; reduced medical costs with co-administration	Mehta et al. [28]

## Data Availability

The data presented in this study are available in this article.

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
