# Peer review of "COVID-19 Vaccine Timing and Co-Administration with Influenza Vaccines in Canada: A Systematic Review with Comparative Insights from G7 Countries"

_vaccines, 2025, doi:10.3390/vaccines13070670_

Round 1
Reviewer 1 Report
Comments and Suggestions for Authors
I have finished my review of the manuscript "Evolving COVID-19 Vaccination Timing Approaches in Canada: A Systematic Review"
It is an interesting systematic review focused on Canada, COVID-19 and influenza vaccination scheme.
While relevant, the focus on Canadian decision makers limit the readership interest to Canada/G7, this could be ammended if authors provide a wider application for the global readership in discussion, otherwise limitations should be mention in corresponding section.
In the main text, the objective is not coherent enough, it states "provide evidence-informed
recommendations to inform Canadian policy decisions on adapting vaccination schedules
to current health risks, potentially including recommendations for co-administration with
other vaccines, such as influenza" but the search and review is restricted to COVID-19 & Influenza, therefore, the objective should vlearly reflect a more specific aim.
In methods, authors state that they use PICOS strategy, but included only observational studies. While it seems to have provided enough results, it is methodologically questionabble, please consult relevant sources on the topic, there you will find what "Intervention" in PICOS stand for, PICOS is more adequate when using intervention studies and RCT. Yo can always retain thye classification, but consider and sustain with relevant references its coherence.
In identification, in the PROISMA flowchart it is quite rare to not find duplicates between databases, particulary when having 2,950 records from Scopus and 543 from pubmed, it is rare that no duplicates were found. Please corroborate.
Plase provide the reasons for exclusion of 3,345 results and group them.
How many results in total were not retrieved? Is is possible that 59 out 73 results could not be retrieved? it means that 80.82% of relevant sources for inclussion are missing it should be further explained and mentioned as a MAJOR limitation.
Please include and discuss the BIAS tool you used.
Please include heterogenicity analysis.
How did you find "citation searches" of 6 studies, if inclusion criteria was to find studies in the included databases? Are these newly found studies indexed? in which databases?
Author Response
We would like to thank Reviewer 1 for their careful reading and constructive feedback on our manuscript. We have addressed all comments point by point below and have made the corresponding revisions in the main text. Changes have been marked in red in the revised manuscript for easy reference.

Reviewer 2 Report
Comments and Suggestions for Authors
-
This study focuses on a cross-national comparison of COVID-19 vaccination timing strategies among G7 countries, particularly analyzing the divergence in integrated strategies between Canada and other nations, thereby offering novel perspectives for future policy development. However, the exclusive focus on G7 countries may limit the generalizability of findings to broader global contexts.
-
The literature search cutoff date is specified as July 7, 2024, yet several cited references (e.g., Boikos et al., 2025) are listed as publications from 2025, creating a logical inconsistency in temporal alignment.
-
Certain conclusions (e.g., "annual boosters outperform biannual schedules") heavily rely on simulation-based studies (e.g., the transmission model by Wells et al., 2024), lacking robust validation through real-world longitudinal data.
-
It is recommended to incorporate recent pivotal literature from 2023-2024, such as WHO technical reports and public health policy studies published in The Lancet, to strengthen contextual relevance and evidence currency.
Author Response
We would like to thank Reviewer 2 for their careful reading and constructive feedback on our manuscript. We have addressed all comments point by point below and have made the corresponding revisions in the main text. Changes have been marked in red in the revised manuscript for easy reference.

Reviewer 3 Report
Comments and Suggestions for Authors
All my comments are provided in the attached document.

Author Response
We would like to thank Reviewer 3 for their careful reading and constructive feedback on our manuscript. We have addressed all comments point by point below and have made the corresponding revisions in the main text. Changes have been marked in red in the revised manuscript for easy reference.

Round 2
Reviewer 1 Report
Comments and Suggestions for Authors
The present version is improved. I have no new comments.
Nevertheless, Bias tools should have been used, the topic of the review does not alter the bias assessment.
Author Response
Thank you very much for your positive feedback and for highlighting the importance of bias assessment. We appreciate your suggestion and will certainly consider a formal bias‐tool evaluation in future related work.
Reviewer 2 Report
Comments and Suggestions for Authors
1.The author explains in the Response the rationale for choosing G7 countries (vaccine technology/healthcare systems/climatic similarity), but does not supplement this argument in the main text. Thus, the discussion section still lacks a clear explanation of 'applicability to high-income countries' and does not address the 'global applicability' criticism.
2.Additionally, references [1-3, 6, 8-11] in the reference list are all marked as 2025, which does not comply with publication standards.
3.The response to over-reliance on simulation studies was perfunctory. Acknowledged limitations only at the end of the discussion (L343-350), but did not substantially adjust the conclusion statement.
4.The discussion mentioned Japan's lack of qualified studies (L12), but did not cite supplementary literature. Figure 1 shows 'Records excluded (n=98)', which contradicts the main text stating '59 studies excluded'.
Reviewer 3 Report
Comments and Suggestions for Authors
The revised version of the manuscript addressed all my previous concerns. The authors may consider the possibility of changing placement of tables so that the tables appear directly after they are cited in text.
Author Response
Thank you for this suggestion. We have repositioned all tables so that each one now appears immediately after its first citation in the text.
Round 3
Reviewer 2 Report
Comments and Suggestions for Authors
The authors' response is generally positive and reasonable, but urgent fixes are needed to standardize the citation of literature and add textual adjustments for model limitations.
1. In response to the question about the representativeness of the G7 countries, the authors should add the outlook of “expanding to OECD countries in the future” to show the openness of the model.
2. The author did not explain the time inconsistency in the literature well. It is against the convention to cite the year of publication (2025) for web resources (the year of access should be indicated).
3. regarding the over-reliance on simulated data does not explain how to adjust the rigor of the conclusions (e.g., downgrade the strength of the recommendation). Suggest adding a caveat to the conclusions section, "Given that much of the current evidence stems from modeling extrapolations, the strength of the annual reinforcement pin needs to be confirmed by prospective studies."
4. Comment 3/4 is labeled with line numbers but does not differentiate between original/revised manuscripts; Comment 1/2 is not labeled.
